# Genome-wide association study identifies multiple new loci associated with Ewing sarcoma susceptibility

Mitchell J. Machiela ⬤ et al.[#]

Ewing sarcoma (EWS) is a pediatric cancer characterized by the *EWSR1-FLI1* fusion. We performed a genome-wide association study of 733 EWS cases and 1346 unaffected individuals of European ancestry. Our study replicates previously reported susceptibility loci at 1p36.22, 10q21.3 and 15q15.1, and identifies new loci at 6p25.1, 20p11.22 and 20p11.23. Effect estimates exhibit odds ratios in excess of 1.7, which is high for cancer GWAS, and striking in light of the rarity of EWS cases in familial cancer syndromes. Expression quantitative trait locus (eQTL) analyses identify candidate genes at 6p25.1 (*RREB1*) and 20p11.23 (*KIZ*). The 20p11.22 locus is near *NKX2-2*, a highly overexpressed gene in EWS. Interestingly, most loci reside near GGAA repeat sequences and may disrupt binding of the EWSR1-FLI1 fusion protein. The high locus to case discovery ratio from 733 EWS cases suggests a genetic architecture in which moderate risk SNPs constitute a significant fraction of risk.

Ewing sarcoma (EWS) is a rare, aggressive pediatric bone or soft-tissue tumor that normally occurs during the second decade of life[1] and likely arises from neural crest- or mesoderm-derived mesenchymal stem cells[2,3]. A translocation between *EWSR1* (22q12) and a member of the ETS transcription factor family, *FLI1* (11q24), in ~85% of cases, is pathognomonic of EWS and provides a distinct and well-defined disease phenotype for genomic characterization[4–6]. The translocation results in an aberrant transcription factor that binds to an ETS-like motif or to GGAA microsatellites and promotes cell transformation through deregulation of target genes responsible for cell cycle control, cell death and migration[6–9]. Aside from *EWSR1-ETS* translocations, there are few other recurrent somatic alterations observed in EWS[10–12].

In 1970, Fraumeni reported a striking disparity in EWS incidence across human populations[13], suggesting an intriguing contribution of germline variation to EWS susceptibility[14]. EWS is predominantly observed in Europeans with an estimated incidence of ~1.5 cases per $10^6$ children and young adults[15]. The estimated incidence in Asian and African populations is substantially lower with annual rates of 0.8 and 0.2 cases per $10^6$ children, respectively, implying genetic variants specific to European ancestry could influence EWS risk[13,15–18]. Despite the rarity of EWS, infrequent and anecdotal instances of familial clustering of EWS in siblings or cousins have also been reported, further suggesting an important genetic component to EWS[19,20]. However, it is notable that EWS is rarely observed in the approximately 120 cancer predisposition syndromes described to date[21].

Our previous genome-wide association study (GWAS) identified susceptibility loci at 1p36.22, 10q21 and 15q15[22]. A follow-up functional study of the 10q21 region localized the association signal to variation in a GGAA microsatellite that, when bound by *EWSR1-FLI1*, functions as an active regulatory element of *EGR2*[23]. Specifically, the A risk allele connected adjacent GGAA repeats by converting an interspaced GGAT motif into a GGAA motif, increasing the number of consecutive GGAA motifs and thus, magnifying the *EWSR1-FLI1*-dependent enhancer activity. Interestingly, *EGR2* knock down inhibits cell proliferation, clonogenicity and tumor growth of EWS cells[23]. Collectively, these findings indicate that germline variation predisposes to EWS risk and can interact with somatically acquired *EWSR1-ETS* fusion proteins to drive carcinogenesis of EWS.

In this report, we perform a GWAS of EWS that combines 401 cases and 682 controls from the previously published EWS GWAS[22] with four new sample sets for a combined total of 733 EWS cases and 1346 unaffected individuals. In total, we investigate EWS associations across 6,876,682 SNPs (genotyped plus high quality imputed) with an overall meta-analysis lambda value of 1.035 (Supplementary Figure 1). We replicate prior associations at 1p36.22, 10q21.3 and 15q15.1 and identify evidence for three new susceptibility loci: 6p25.1, 20p11.22 and 20p11.23 (Table 1, Fig. 1, Supplementary Table 1, Supplementary Figures 2–3).

**Table 1 Magnitude and strength of association for previously published and new EWS susceptibility loci**

| Region | Top SNP | Ref | Risk | Odds Ratio | 95% Confidence Interval | | Assoc *P*-value | Het *P*-value |
|--------|---------|-----|------|-----------|-------|-------|---------------|-------------|
| 1p36.22 | rs113663169 | C | T | 2.05 | 1.71 | 2.45 | 4.32E-15 | 0.58 |
| 6p25.1 | rs7742053 | C | A | 1.80 | 1.48 | 2.18 | 2.78E-09 | 0.12 |
| 10q21.3 | rs10822056 | C | T | 1.76 | 1.54 | 2.02 | 1.92E-16 | 0.45 |
| 15q15.1 | rs2412476 | C | T | 1.73 | 1.48 | 2.01 | 1.45E-12 | 0.93 |
| 20p11.22 | rs6047482 | T | A | 1.74 | 1.49 | 2.04 | 2.55E-12 | 0.90 |
| 20p11.23 | rs6106336 | T | G | 1.74 | 1.43 | 2.12 | 2.33E-08 | 0.16 |

*Ref* reference allele, *Risk* risk allele, *RAF* risk allele frequency (CEU), *Assoc P-value* Meta-analysis combined association *P*-value for GWAS discovery set, *Het P-value* *P*-value from test of heterogeneity for GWAS discovery set

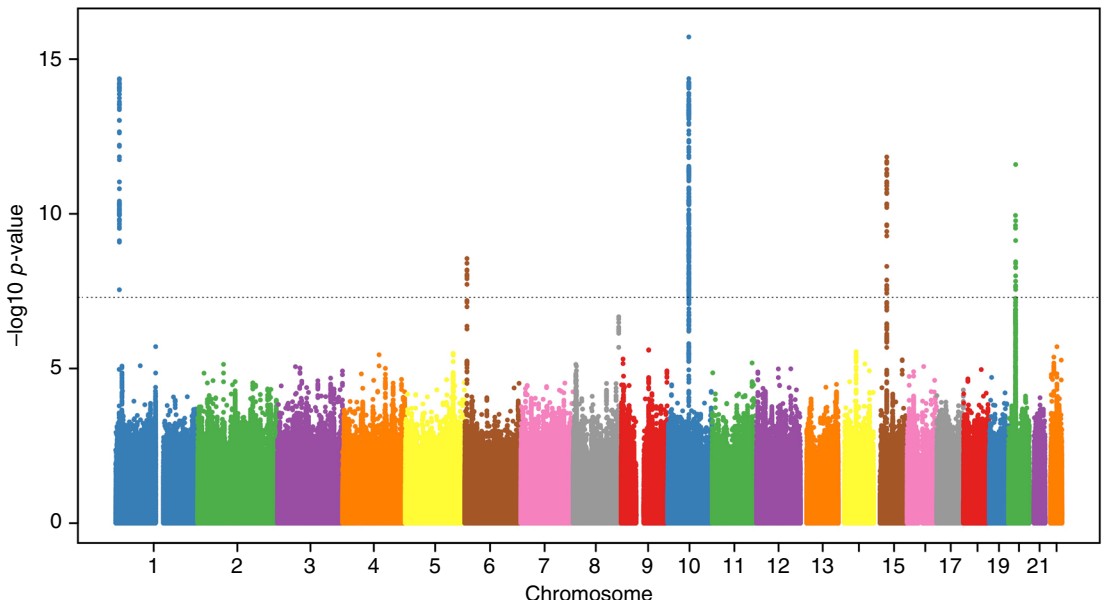

**Fig. 1** Manhattan plot of meta-analysis $-\log_{10}$ *P*-values for the association of each SNP with EWS risk. Association -log10 *p*-values for each tested genetic variant are plotted. Chromosomes are plotted sequentially across the x-axis with the scale proportional to chromosomal size. Colors are used to visualize differences in chromosome. The dotted line indicates genome-wide significance ($P<5\times10^{-8}$)

## Results and discussion

**Methods Summary.** Our analysis was restricted to individuals of >80% estimated European ancestry based on a principal component analysis of population substructure. EWS cases were confirmed by medical record review, which included checking for the presence of *EWSR1-ETS* fusions when data was available. Principal component matching was performed to select a genetically homogeneous set of adult controls who were cancer-free as of age 50 for each EWS case. Sample and SNP quality control exclusions were carried out to ensure unrelated, high quality samples for association analysis with accurate genotype assays. Missing genotypes were imputed using 1000 Genomes Phase 3 haplotypes as a reference[24]. We combined results across studies using a fixed effects meta-analysis. Variants with minor allele frequencies <5% or significant evidence for heterogeneity were filtered from the final results. A more detailed description of our experimental methods and analysis technique is available in Methods.

**Replication of prior EWS GWAS.** Our analysis provided strong replication of three previously discovered EWS susceptibility loci[22] and aided in refining the association signals. We observed rs113663169 as the most significant variant tagging the 1p36.22 locus (OR = 2.05, 95% CI = 1.71–2.45, $P$-value$_{meta}$ = 4.32×10$^{-15}$). This variant is in high linkage disequilibrium (LD) with the original reported variant rs9430161 ($R^2_{CEU}$ = 0.97, $D'_{CEU}$ = 1.00;[25] OR = 2.03, 95% CI = 1.70–2.42, $P$-value$_{assoc}$ = 6.3×10$^{-15}$)[22] and is located upstream of *TARDBP*, a transcriptional repressor that shares structural similarities with *EWSR1* and binds RNA regulatory elements. At the 10q21 locus, we observed rs10822056 with the strongest association (OR = 1.76, 95% CI = 1.54–2.02, $P$-value$_{meta}$ = 1.92×10$^{-16}$). This variant is correlated with the reported variant from the original GWAS, rs224278 ($R^2_{CEU}$ = 0.52, $D'_{CEU}$ = 0.92;[25] OR = 1.71, 95% CI = 1.49–1.96, $P$-value$_{assoc}$ = 6.9×10$^{-15}$)[22], as well as the putatively functional variant, rs79965208 ($R^2_{CEU}$ = 0.24, $D'_{CEU}$ = 0.57;[25] OR = 1.42, 95% CI = 1.24–1.63, $P$-value$_{assoc}$ = 5.3×10$^{-7}$)[23]. Interestingly, as indicated previously[23], the conditional analysis at 10q21.3 suggests evidence for a residual independent signal in this region, although larger EWS GWAS are needed to confirm the presence of multiple independent signals. Finally, at 15q15.1 we observed rs2412476, a tagging variant strongly associated with EWS (OR = 1.73, 95% CI = 1.48–2.01, $P$-value$_{meta}$ = 1.45×10$^{-12}$). This variant is in moderate LD with rs4924410 from the original GWAS ($R^2_{CEU}$ = 0.18, $D'_{CEU}$ = 1.00[25]; OR = 1.62, 95% CI = 1.41–1.86, $P$-value$_{assoc}$ = 5.4×10$^{-12}$)[22] and is located near several genes including *BMF*, *BUB1B* and *PAK6*.

**Newly identified EWS susceptibility loci.** Our analysis identified suggestive evidence for novel genomic associations ($P$-value$_{meta}$ < 5 × 10$^{-7}$) in four genomic regions (Table 1, Supplementary Table 1): 6p25.1, 8q24.23, and 20p11.22 and 20p11.23. To validate signals from imputed variants in these regions, we performed allele-specific TaqMan PCR for a subset of 335 GWAS samples on the following variants: rs7744366 (6p25.1), rs7832583 (8q24.23), rs12106193 (20p11.22) and rs6106336 (20p11.23). All PCR-validated genotypes had over 93–99% concordance with imputed genotypes indicating high accuracy of imputation in these regions (Supplementary Table 2). Additionally, these signals were replicated in two independent series of EWS cases and controls: a European set from the Institute Curie containing of 480 EWS cases and 576 controls[22], and a German set from LMU Munich containing 177 EWS cases and 3502 controls. All combined association $P$-values (GWAS discovery+independent replication sets) were below genome-wide significance levels ($P$-value$_{meta}$ < 5 × 10$^{-8}$, Supplementary Table 1, Supplementary Figures 2–5) except for the 8q24.23 locus ($P$-value$_{meta}$ = 1.44 × 10$^{-7}$). The 6p25.2 and 20p11.22 signals were independently replicated in both German and European replication sets; however, the 8q24.23 signal was only significant in the European set ($P$-value$_{assoc}$ = 0.007) and the 20p11.23 signal was only replicated in the German set ($P$-value$_{assoc}$ = 0.036).

**EWS susceptibility locus at 6p25.** We identified a new locus on 6p25.1 tagged by rs7742053 (OR = 1.80, 95% CI = 1.48–2.18, $P$-value$_{meta}$ = 2.78×10$^{-9}$) with the A allele being the risk associated allele (Supplementary Table 3). The marker variant rs7742053 is telomeric to *RREB1*, *SSR1* and *CAGE1*. Expression quantitative trait locus (eQTL) analysis using rs1286037, a correlated surrogate for rs7742053 ($R^2_{CEU}$ = 0.49, $D'_{CEU}$ = 1.00)[25], identified allele specific expression differences in *RREB1*, with the risk A allele of rs7742053 corresponding to increased levels of *RREB1* expression ($P$-value$_{Wald}$ = 0.01, Table 2). *RREB1* encodes the RAS responsive element (RRE) binding protein 1, a zinc-finger transcription factor that binds to RRE in gene promoters[26]. *RREB1* is expressed in EWS tumors at higher levels than other pediatric sarcomas (Supplementary Figure 6), suggesting regulation of *RREB1* may be particularly important for EWS. In addition, the 6p25.1 locus shows evidence for an interaction between germline

---

### Table 2 Functional associations for newly identified EWS susceptibility loci

| Locus | SNP | Risk Allele | Gene | eQTL Proxy SNP | Proxy R2/D′ | eQTL P-value | eQTL Direction | EWSR1-FLI1 knock down |
|-------|-----|-------------|------|----------------|-------------|--------------|----------------|------------------------|
| 6p25.1 | rs7742053 | A | *CAGE1* | rs1286037 | 0.49/1.00 | 0.939 | — | |
| | | | *LY86* | rs1286037 | 0.49/1.00 | 0.727 | — | |
| | | | *LY86-AS1* | rs1286037 | 0.49/1.00 | 0.487 | — | |
| | | | *RREB1* | rs1286037 | 0.49/1.00 | 0.010 | ↑ | ↓ |
| | | | *SSR1* | rs1286037 | 0.49/1.00 | 0.630 | — | |
| 8q24.23 | rs7832583 | C | none | — | — | — | — | |
| 20p11.22 | rs6047482 | A | *KIZ* | rs6137387 | 0.60/1.00 | 0.478 | — | ↓ |
| | | | *NKX2-2* | rs6137387 | 0.60/1.00 | 0.127 | — | ↓ |
| | | | *PAX1* | rs6137387 | 0.60/1.00 | 0.489 | — | |
| | | | *XRN2* | rs6137387 | 0.60/1.00 | 0.277 | — | |
| 20p11.23 | rs6106336 | G | *KIZ* | rs6047241 | 1.00/1.00 | 0.014 | ↑ | ↓ |
| | | | *NKX2-2* | rs6047241 | 1.00/1.00 | 0.359 | — | ↓ |
| | | | *XRN2* | rs6047241 | 1.00/1.00 | 0.260 | — | |

Risk allele is the allele associated with increased EWS risk. eQTL $P$-value is from a Wald test of the genotype beta value. eQTL direction is the effect the risk allele has on quantitated gene expression. EWSR1-FLI1 knock down indicates the effect of EWSR1-FLI1 knock down on relative gene expression. The up arrow (↑) indicates increased expression and the down arrow (↓) indicates decreased expression

---

variation and *EWSR1-FLI1* fusion proteins. ChIP-seq of acety-lated H3K27 (H3K27ac) indicates an area of open chromatin that spans a polymorphic GGAA microsatellite near rs7742053 (Supplementary Figure 7-8, Supplementary Tables 4-5). ChIP-seq analysis of *EWSR1-FLI1* in the A673 and TC-71 EWS cell lines confirm *EWSR1-FLI1* binding to this GGAA microsatellite at 6p25.1. Further, knock down of *EWSR1-FLI1* in xenografts derived from the A673/TR/shEF1 EWS cell line results in strong downregulation of *RREB1* in vivo (Supplementary Figure 9). Several variants correlated with rs7742053 are in contiguity with the GGAA repeat and may be candidate functional variants that disrupt *EWSR1-FLI1* binding (Supplementary Table 5). One such variant, rs10541084, a -/GAAG indel is located at the telomeric end of the nearest GGAA microsatellite, is in LD with rs7742053 ($R^2_{CEU} = 0.15$, $D'_{CEU} = 0.92$)[25], and is nominally associated with EWS (OR = 1.20, 95% CI = 1.04–1.37, $P\text{-value}_{meta} = 0.01$). Interestingly, the rs7742053 risk A allele is correlated with the rs10541084 GAAG allele which is more common in Europeans, extends the microsatellite GGAA repeat sequence, and could enhance binding of *EWSR1-FLI1*. This evidence suggests that a similar mechanism as in the 10q21 locus[23] may be acting at the 6p25.1 locus in which variation of a GGAA repeat affects *EWSR1-FLI1* binding leading to altered expression of *RREB1* or an alternative nearby gene. Further functional work at 6p25.1 is required to clarify which variants are functionally responsible for the susceptibility signal.

**EWS susceptibility locus at 20p11.** We identified an association signal spanning chromosome 20p11.22-23. The strongest asso-ciation signal was on 20p11.22 tagged by rs6047482 (OR = 1.74, 95% CI = 1.49–2.04, $P\text{-value}_{meta} = 2.55 \times 10^{-12}$). The A allele is the risk allele with a higher frequency observed in 1000 Genome Europeans than in Africans (Supplementary Table 3). While no statistically significant eQTL was observed between this locus and nearby genes (Table 2), the nearest transcript, *NKX2-2*, is of high interest; *NKX2-2*, NK2 homeobox 2, encodes a homeobox domain protein that is a likely nuclear transcription factor, which is overexpressed in the presence of *EWSR1-FLI1* fusions in EWS tumors[27,28]. Our analysis did not detect significant allele specific expression differences for *NKX2-2* in association with rs6047482 (eQTL $P\text{-value}_{Wald}$ with rs12106193 = 0.17, $R^2_{CEU}$ and $D'_{CEU}$ between rs6047482 and rs12106193 = 0.67 and 1.00, respectively)[25]. We explored eQTLs for other tissue types in GTEx with surrogate SNPs in moderate to high linkage dis-equilibrium with rs6047482, but found no evidence for an eQTL with *NKX2-2* in these tissues likely due to EWS specific expres-sion of *NKX2-2* (Supplementary Table 6)[29]. It is plausible that *EWSR1-FLI1*-induced elevated *NKX2-2* expression levels in EWS cells hamper our ability to detect allele specific expression pat-terns of *NKX2-2* that may be important for EWS transformation in the EWS progenitor cells. Further eQTL analyses in a large set of mesenchymal stem cells, the suspected EWS cell-of-origin, should enable this hypothesis to be tested. As with the 6p25.1 locus, ChIP-seq data show that *EWSR1-FLI1* binds to one or more polymorphic GGAA microsatellites proximal to the tagging variants (Supplementary Figure 10) suggesting that variation in this region could exert an effect through *NKX2-2* gene regulation in EWS progenitors and in turn through *EWSR1-FLI1* binding in EWS cells. Importantly, the six lead SNPs are on average significantly closer to *EWSR1-FLI1* bound elements than would be expected by chance on a chromosome-wide level ($P\text{-values}_{Wilcoxon} = 0.0025$ and $0.0009$ in A673 and TC71 cell lines, respectively) (Supplementary Figure 8, Supplementary Table 4).

**Independent EWS susceptibility signal at 20p11.** In the search for additional independent loci at each EWS susceptibility locus (Supplementary Figure 4), we identified a second, independent signal on 20p11.23 tagged by rs6106336 based on a conditional analysis using the discovery marker, rs6047482 ($R^2_{CEU} = 0.003$, $D'_{CEU} = 0.23$; OR = 1.74, 95% CI = 1.43–2.12, $P\text{-value}_{meta} = 2.33 \times 10^{-8}$, $P\text{-value}_{conditional} = 5.2 \times 10^{-8}$, Fig. 2) with the G allele acting as the risk associated allele. A distinct eQTL was observed between a highly correlated surrogate for rs6106336, rs6047241 ($R^2_{CEU} = 1.00$, $D'_{CEU} = 1.00$), and *KIZ*, kizuna centrosomal protein, (also known as *PLK1S1*) with the risk G allele associated with increased expression ($P\text{-value}_{Wald} = 0.01$, Table 2). This eQTL at 20p11.23 with *KIZ* does not appear to be restricted to EWS and was observed in other GTEx tissues (e.g., artery, sun-exposed skin, testis and whole blood; Supplementary Table 6). *KIZ* localizes to the centrosomes and functions to strengthen and stabilize the pericentriolar region prior to spindle formation[30]. While limited evidence suggests *EWSR1-FLI1* binding in this region, H3K27ac patterns suggest areas of open chromatin that may harbor variants important for regulation of nearby gene products (Supplementary Figure 11).

**EWS genetic risk score.** In light of the observed set of EWS loci, all with high estimated effect sizes, we generated a genetic risk score (GRS) combining risk alleles from the six EWS suscep-tibility loci to test the ability of an EWS GRS to discriminate between EWS cases and cancer-free adult controls (Supplemen-tary Figure 12). On average, EWS cases carried 1.08 more risk alleles than controls (7.08 average risk alleles in EWS cases, 6.01 average risk alleles in controls; $P\text{-value}_{T\text{-test}} = 2.44 \times 10^{-63}$). Due to the rarity of EWS and the relatively high frequency of these common susceptibility alleles, absolute risks of EWS associated with these six EWS susceptibility loci are low suggesting population-based screening using these six variants is unlikely to be effective.

**Genetic architecture of EWS.** Our new, expanded GWAS of Ewing sarcoma has identified three new loci and also validated the three previously reported susceptibility regions. In analyses of the new loci, there is evidence of informative eQTLs with nearby biologically plausible candidate genes that could be likely target genes for future functional investigations. Additionally, *EWSR1-FLI1* ChIP-seq data suggest evidence for potential interactions of germline variation at the 6p25.1 and 20p11.22 loci with the *EWSR1-FLI1* fusion protein as recently discovered at the 10q21 locus[23]. It is remarkable that six independent susceptibility regions with relatively large effect sizes (estimated OR > 1.7) have been discovered in a sample of 733 EWS cases. These results provide a strong contrast to GWAS findings for the vast majority of cancers that report estimated effect sizes less than 1.2. Inter-estingly, GWAS in two highly heritable cancers (e.g., testicular and thyroid)[31,32] have also identified susceptibility alleles with effect sizes in the range of what is observed for Ewing sarcoma. The efficiency of our discovery as well as the higher estimated EWS odds ratios could be related to the lack of tumor hetero-geneity in our Ewing sarcoma GWAS, because most EWS cases studied had a pathologically confirmed *EWSR1-ETS* fusion, a pathognomonic molecular feature of the EWS diagnosis. Fur-thermore, our results suggest the underlying EWS genetic sus-ceptibility architecture harbors a substantial number of moderate effect common variants, which is striking because Ewing sarcoma has not been considered to be highly heritable. In conclusion, our study provides support for a strong inherited genetic component to EWS risk and suggests interactions between germline variation

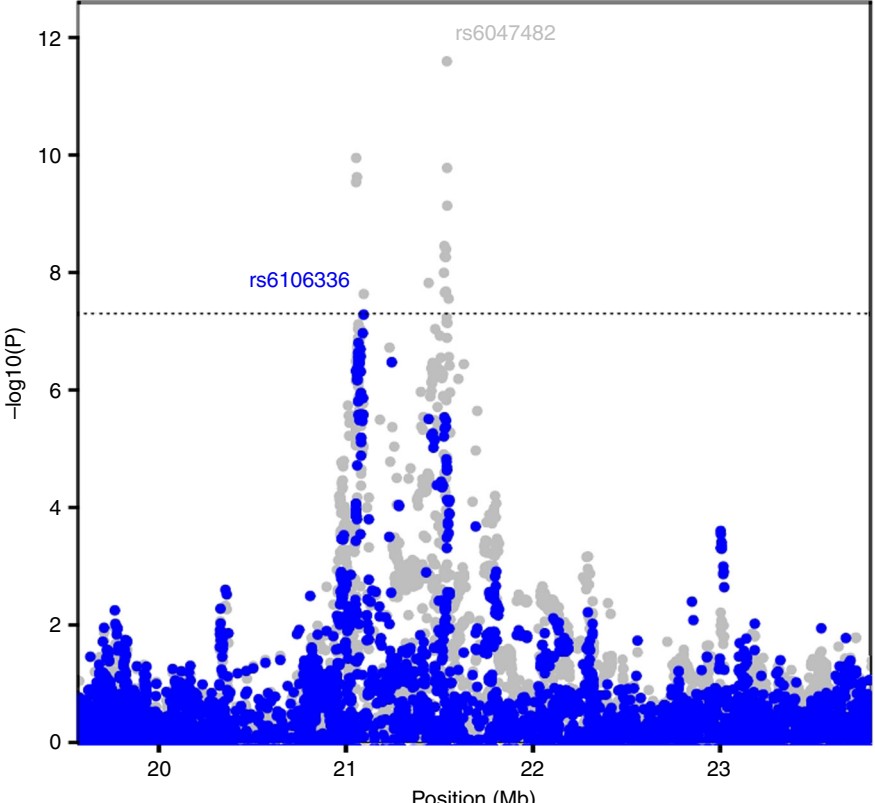

**Fig. 2** Conditional analysis at the 20p11.22-23 region. Overall meta-analysis –log$_{10}$ P-values are plotted in gray in the background. In the foreground, meta-analysis –log$_{10}$ p-values when the top tagging SNP is the region (rs6047482) is conditioned on is plotted in blue. A second independent signal, tagged by rs6106336, remains

and somatically acquired *EWSR1-FLI1* translocations are important etiologic contributors to EWS risk.

## Methods

**Study populations**. EWS cases and controls for this GWAS originated from several contributing studies. A set of published French EWS cases ($N = 401$) and ancestry matched controls ($N = 682$) was extracted from the previously published GWAS on EWS[22]. In addition, we combined a set of 122 French EWS cases from the Institut Curie, 19 EWS cases from the National Cancer Institute (NCI) Center for Cancer Research (CCR), 29 EWS cases from the NCI Bone Disease and Injury Study[33] along with 162 EWS cases from the Childhood Cancer Survivor Study (CCSS)[34,35]. The SNPWEIGHTS software was used to calculate the percentage of European ancestry using a set of population inference SNPs[36,37]. Only EWS cases with >80% genetically estimated European ancestry were included in the analysis. This resulted in a total of 749 EWS cases combined with 682 cancer-free controls from the original EWS GWAS. To increase the sample size of available controls, we identified a set of adult controls previously genotyped at the NCI Division of Cancer Epidemiology and Genetics who were cancer-free at age 50, of European ancestry and genotyped on a current generation of high-density Illumina genotyping platform. Controls originated from the Prostate Lung Colorectal and Ovarian Cancer Screening Trial ($N = 419$), American Cancer Society Cancer Prevention Study II ($N = 171$), and the Spanish Bladder Cancer Study ($N = 74$). EWS cases without available controls were split into two groups: (1) the CCSS group and (2) a Curie/NCI group that contained EWS cases from the Institute Curie, NCI CCR and NCI Bone Disease and Injury Study. Principal component matching was performed to identify ancestry matched controls for each EWS case that were close genetic matches (Supplementary Figure 13). The first three principal components were used as matching factors and a 2:1 matching ratio of controls to cases was carried out based on the availability of close control matches. Matching was performed first for the CCSS set to maximize matches with available controls on high-density arrays. In total, our final analysis set contained 733 EWS cases and 1346 cancer-free controls.

All EWS cases were confirmed by medical record review and the presence of a specific *EWSR1-ETS* translocation were noted when data was available. Adult controls were of European ancestry and cancer-free at time of DNA collection. Each participant provided informed consent and each participating study was approved by the Institutional Review Boards of their study center.

**Genome-wide SNP genotyping and quality control assessment**. Samples from the previously published EWS GWAS[22] were derived from bone marrow, blood or tumor tissue. Genomic DNA was isolated using proteinase K lysis and phenol chloroform extraction method. Genome-wide genotyping was performed on Illumina 610 Quadv1 arrays at Integragen (Evry, France). For the CCSS EWS cases, DNA was extracted using standard methods from blood, saliva (Oragene), or buccal cells. For CCSS EWS cases with insufficient DNA, whole genome amplification was performed[38]. Genotyping of CCSS EWS cases and quality control replicates was conducted at the NCI Cancer Genomics Research Laboratory (CGR) on the HumanOmni5Exome array. Genotypes were called using default parameters in GenomeStudio (Illumina).

All de novo genotyping of EWS cases was performed at the NCI CGR on the Illumina OmniExpress-24 v1.1 array. Genotyping was performed according to manufacturer's guidelines using the Infinium HD Assay automated protocol. For each sample, 400 ng of input DNA was denatured and neutralized then isothermally amplified by whole-genome amplification. The amplified product was enzymatically fragmented, then precipitated and re-suspended before hybridization to the BeadChip. Single-base extension of the oligos on the BeadChip, using the captured DNA as a template, incorporated tagged nucleotides on the BeadChip, which were subsequently fluorophore labeled during staining. An Illumina iScan scanned the BeadChips at two wavelengths to create image files. Genotypes were called using default parameters in GenomeStudio. Standard quality control checks were performed to ensure included EWS cases had high genotype completion rates (≥95%), sex concordance, normal rates of heterozygosity and no unexpected duplicates or cases of high relatedness (IBD < 0.1).

Missing genotypes were imputed in three sets: (1) the previously published GWAS set[22], (2) the CCSS EWS cases and matched controls and (3) the Curie/NCI EWS cases and matched controls. For sets 2 and 3, only the common set of shared genotypes between EWS cases and cancer-free controls was used as input for imputation. All samples were first phased using SHAPEIT[39] and subsequently imputed in IMPUTE2[40] using the 1000 Genomes Phase 3 release as the reference[24]. Only SNPs with study info score greater than 0.3 and study minor allele counts greater than or equal to 5 were carried on into the association analysis.

**Statistical analysis**. Association analyses were performed individually in each of the three imputation sets using SNPTEST. Statistical adjustment was carried out for principal components (PCs) used for matching (in the CCSS and Curie/NCI sets) and those PCs significantly associated with EWS risk. PCs were calculated in PLINK[41] using a set of population inference SNPs[36]. In the previously published

set, statistical adjustment was carried out for PC 2, PC 3, PC 6, PC 11 and PC 12. In the CCSS set, PC 1, PC 2, PC 3, PC 5, PC 17 and PC 20 were adjusted for. For the Curie/NCI set, statistical adjustment was carried out for PC 1, PC 2, PC 3, PC 4, PC 9, PC 15, PC 17 and PC 18.

Resulting statistical associations from each of the three sets were combined in PLINK by fixed effects meta-analysis. When performing the meta-analysis, beta estimates and standard errors corresponded to the same reference alleles across all three sets were merged. Resulting associations from the meta-analysis were filtered to include single nucleotide variants with minor allele frequency greater than 0.05 and P-values for Cochrane's Q heterogeneity tests greater than 0.01.

Genetic risk scores (GRS) of the three previously discovered and three new independent loci were calculated to investigate the ability of tagging variants from these six loci to discriminate between EWS cases and controls. GRS were constructed by summing the number of risk alleles an individual carried across all six EWS loci. A T-test was used to assess statistical differences in mean risk alleles carried in EWS cases as compared to controls. Area under the receiver operating characteristic curve (AUC) was the metric used to measure discriminative ability of the GRS.

**Genotype validation**. For each of the four newly discovered EWS loci, a top genome-wide significant SNP was genotyped by TaqMan in a subset of 335 samples to validate signals from imputation. At each locus, genotyping was attempted for the top associated SNP. When assay design failed for the top tagging SNP, the next most highly SNP was sequentially attempted until an appropriate assay design was established. The following SNPs were genotyped by TaqMan for each locus: rs7744366 (6p25.1), rs7832583 (8q24.23), rs12106193 (20p11.23) and rs6106336 (20p11.23). Standard protocols were followed according to manufacturer's guidelines when performing the TaqMan assays (Supplementary Table 7). Assays were first tested on HapMap samples to ensure validity before testing on EWS cases and controls included in the GWAS.

**Replication study**. An independent set from the Institute Curie and European collaborators consisting of 480 EWS cases and 576 controls and an independent German set from LMU Munich containing 177 EWS cases and 3502 population-based controls from the KORA S4 study were used as a replication sets to confirm associations at the 6p25.1, 8q24.23, 20p11.22 and 20p11.23 regions. The following SNPs were genotyped by TaqMan for each locus: rs7744366 (6p25.1), rs7832583 (8q24.23), rs12106193 (20p11.23) and rs6106336 (20p11.23) (Supplementary Table 7). Standard manufacturer's protocols were followed when performing allele-specific PCRs with these TaqMan assays. The 3502 German controls from the KORA S4 cohort were genotyped with the Affymetrix Axiom array and imputed using the 1000 Genomes phase 3 as well as the Haplotype Reference Consortium (HRC) reference panels.

**EWS functional data**. All expression quantitative trait loci (eQTL) analyses were performed using previously published expression data from 117 EWS samples[22]. Samples were profiled using Affymetrix Human Genome U133 Plus 2.0 gene expression arrays. Expression data are publically available at the Gene Expression Omnibus (GEO) web portal (GSE34620). Affymetrix expression data were normalized with the NormalizeBetweenArrays function of the LIMMA package (http://web.mit.edu/~r/current/arch/i386_linux26/lib/R/library/limma/html/normalizebetweenarrays.html). Wald tests of estimated betas from linear regression models were performed to asses for allele specific differences in gene expression levels.

**Chromatin immunoprecipitation (ChIP)**. ChIP experiments were performed following manufacture instructions using iDeal ChIP-seq kit for transcription factors and for histones (Diagenode) with respectively a rabbit polyclonal anti-FLI1 antibody (Ab15289, Abcam) and a rabbit polyclonal anti-H3K27ac (Ab4729, Abcam). The Ewing sarcoma A673 cell line was obtained from the American Type Culture Collection (ATCC) and the Ewing sarcoma TC-71 cell line was obtained from the German Collection of Microorganisms and Cell Cultures (DSMZ). STR-profiling proved each cell line matched with the reference profile provided by ATCC and DSMZ, respectively; and cells were routinely tested for mycoplasma contamination by PCR. Briefly, the EWS cell lines were fixed for 10 min with 1% of methanol-free formaldehyde (28908, Thermo-Scientific). Chromatin was sonicated (Bioruptor, Diagenode) for 20 cycles (30-sec on, 30-sec off) set at position "high" to generate DNA fragments with an average size around 150–300pb. For ChIP sequencing, the libraries were generated using TruSeq ChIP library preparation kit (Illumina) and sequenced on Illumina HiSeq 2500 (single end, 100 bp). Reads were aligned to hg19 reference genome with Bowtie2[42]. Peaks were called with MACS2[43] with the option narrow for FLI1 ChIP-seq and broad for H3K27ac ChIP-seq. For each cell line, ChIP-seq results were normalized according to their input sample.

**Comparison of gene expression levels across cancer types**. Publically available gene expression data for 19 different cancer entities comprising in total 616 tumor samples, which were all profiled on Affymetrix Human Genome U133 Plus 2.0 gene expression arrays, were downloaded from the GEO or the Array Express platform of the European Bioinformatics Institute (EBI) (accession codes:

GSE68015, GSE13433, GSE32569, GSE19404, GSE35493, GSE58697, GSE34620, GSE34800, GSE60740, GSE19348, GSE17743, GSE8167, GSE53224, GSE16476, E-MEXP-3628, GSE14827, GSE33458, E-TABM-1202, GSE29683, GSE20196, GSE21050). Microarray data were simultaneously normalized using Robust Multi-Chip Average (RMA)[44] using brainarray CDF (v19 ENTREZG) yielding one optimized probe-set per gene[45].

**Analysis of EWSR1-FLI1-modulated genes in vivo**. For transcriptome-wide analysis of EWSR1-FLI1-modulated genes $5 \times 10^6$ A673/TR/shEF1 cells[46], which contain a doxycycline (dox)-inducible shRNA against EWSR1-FLI1, were injected subcutaneously in the flanks of immunocompromised NSG (Nod scid gamma) mice. A673/TR/shEF1 cells were authenticated by STR-profiling to match their parental A673 cell line and were routinely tested for mycoplasma contamination by PCR. When tumors reached an average volume of 180 mm³, mice were randomized and either received 2 mg/l dox (Sigma) and 5% sucrose in the drinking water (dox+) or only 5% sucrose (dox –). Mice were sacrificed 96 h after beginning of dox-treatment, and tumors were collected for RNA and histological analysis. Total RNA was extracted using the ReliaPrep miRNA Cell and Tissue Miniprep System (Promega). Knock down of EWSR1-FLI1 was confirmed by qRT-PCR as described[23], and proved to be downregulated onto 15% of the control (sucrose only). Routine histology (H&E stains) of the tumors confirmed high tumor purity (>95%). The transcriptomes of 3 dox(+) and 3 dox (−) were profiled on Affymetrix Clariom D arrays (RIN > 9). Microarray data were simultaneously normalized on gene level using Signal Space Transformation Robust Multi-Chip Average (SST-RMA) and Affymetrix CDF. Animal experiments were conducted in accordance with the recommendations of the European Community (86/609/EEC), the Government of Upper Bavaria (Germany), and UKCCCR (guidelines for the welfare and use of animals in cancer research). The sample size was not pre-determined and no blinding was performed.

**Data availability**. Data from the newly genotyped individuals in EWS GWAS is available on dbGaP under accession number phs001549.v1.p1 Data from CCSS is available on dbGaP under accession number phs001327.v1.p1.

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

## Acknowledgements

This work was supported by the Intramural Research Program of the U.S. National Cancer Institute and the Intramural Research Program of the American Cancer Society. This work was supported by grants from the Institut Curie, the Inserm, the Ligue Nationale Contre le Cancer (Equipe labellisée, Carte d'Identité des Tumeurs program and Recherche Epidémiologique 2009 program), the ANR-10-EQPX-03 from the Agence Nationale de la Recherche, the European PROVABES (ERA-649 NET TRANSCAN JTC-2011), and ASSET (FP7-HEALTH-2010-259348) projects. This research was supported by FP7 grant "EURO EWING Consortium" No. 602856 and the following associations: Courir pour Mathieu, Dans les pas du Géant, Les Bagouzamanon, Enfants et Santé, M la vie avec Lisa, Lulu et les petites bouilles de lune, les Amis de Claire, l'Etoile de Martin and the Société Française de lutte contre les Cancers et les leucémies de l'Enfant et de l'adolescent. The laboratory of T. G. P. Grünewald is supported by grants from the 'Verein zur Förderung von Wissenschaft und Forschung an der Medizinischen Fakultät der LMU München (WiFoMed)', by LMU Munich's Institutional Strategy LMU excellent within the framework of the German Excellence Initiative, the 'Mehr LEBEN für kreb-skranke Kinder—Bettina-Bräu-Stiftung', the Walter Schulz Foundation, the Wilhelm Sander-Foundation (2016.167.1), and by the German Cancer Aid (DKH-111886 and DKH-70112257). D. Surdez is supported by SiRIC (Grant « INCa-DGOS-4654). We thank the following clinicians for providing samples used in this study: C. Alenda, F. Almazán, D. Ansoborlo, L. Aymerich, L. Benboubkher, C. Beléndez, C. Berger, C. Ber-geron, P. Biron, J.Y. Blay, E. Bompas, H. Bonnefoi, P. Boutard, B. Bui-Nguyen, D. Chauveaux, C. Calvo, A. Carboné, C. Clement, T. Contra, N. Corradini, A.S. Defachelles, V. Gandemer-Delignieres, A. Deville, A. Echevarria, J. Fayette, M. Fraga, D. Frappaz, J.L. Fuster, P. García-Miguel, J.C. Gentet, P. Kerbrat, V. Laithier, V. Laurence, P. Leblond, O. Lejars, R. López-Almaraz, B. López-Ibor, P. Lutz, J.F. Mallet, L. Mansuy, P. Marec Bérard, G. Margueritte, A. Marie Cardine, C. Melero, L. Mignot, F. Millot, O. Minckes, G. Margueritte, C. Mata, M.E. Mateos, M. Melo, C. Moscardó, M. Munzer, B. Narciso, A. Navajas, D. Orbach, C. Oudot, H. Pacquement, C. Paillard, Y. Perel, T. Philip, C. Piguet, M.I. Pintor, D. Plantaz, E. Plouvier, S. Ramirez-Del-Villar, I. Ray-Coquard, Y. Reguerre, M. Rios, P. Rohrlich, H. Rubie, A. Sastre, G. Schleiermacher, C. Schmitt, P. Schneider, L. Sierrasesumaga, C. Soler, N. Sirvent, S. Taque, E. Thebaud, A. Thyss, R. Tichit, J.J. Uriz, J. P. Vannier, F. Watelle-Pichon. This work was supported by the Instituto de Salud Carlos III (PI16CIII/00026) and the Asociación Pablo Ugarte, Fundación Sonrisa de Alex, ASION-La Hucha de Tomás, Sociedad Española de Hematología y Oncología Pediátricas. The Childhood Cancer Survivor Study is supported by the National Cancer Institute (CA55727, G.T. Armstrong, Principal Investigator), with funding for genotyping from the Intramural Research Program of the National Institutes of Health, National Cancer Institute. The KORA study was initiated and financed by the Helmholtz Zentrum München—German Research Center for Environmental Health, which is funded by the German Federal Ministry of Education and Research (BMBF) and by the State of Bavaria. Furthermore, KORA research was supported within the Munich Center of Health Sciences (MC-Health), Ludwig-Maximilians-Universität, as part of LMUinnovativ.

## Author contributions

M.J.M., S.J.C. and O.D. designed the study. M.J.M., T.G.P.G., S.J.C. and O.D. wrote the manuscript. M.J.M., T.G.P.G., O.M., S.G.-L. and F.T. performed the statistical analysis. R. A.R., C.L.D., L.B., K.J., M.M., K.W., W.Z. and M.Y. performed the genotyping and quality control. The following authors provided samples and/or data to the study and commented on the manuscript: T.G.P.G., D.S., S.R., O.M., E.K., R.A.R., S.Z., S.G.-L., S.Ballet, E.L., V.L., J.M., G.P., H.K., N.G., U.K., A.G.-N., P.P., J.A., A.P.-G., N.C., P.M.B., N.D.F., N.R., D.G.C., R.N.H., J.Khan, G.T.A., W.M.L., S.Bhatia, L.L.R., A.E.K., J.Kriebel, T.M., M.Metzler, W.H., K.S., T.K., U.D., L.M.M., L.M., M.A.T., and O.D.

## Additional information

Mitchell J. Machiela [1], Thomas G.P. Grünewald [2,3,4], Didier Surdez [5,6], Stephanie Reynaud[6,7], Olivier Mirabeau[5,6], Eric Karlins[1,8], Rebeca Alba Rubio[2], Sakina Zaidi[5,6], Sandrine Grossetete-Lalami[5,6], Stelly Ballet[6,7], Eve Lapouble[6,7], Valérie Laurence[6], Jean Michon[6], Gaelle Pierron [6,7], Heinrich Kovar [9], Nathalie Gaspar[10], Udo Kontny[11], Anna González-Neira[12], Piero Picci[13], Javier Alonso [14], Ana Patino-Garcia[15], Nadège Corradini[16], Perrine Marec Bérard[16], Neal D. Freedman[1], Nathaniel Rothman[1], Casey L. Dagnall [1,8], Laurie Burdett[1,8], Kristine Jones[1,8], Michelle Manning[1,8], Kathleen Wyatt[1,8], Weiyin Zhou[1,8], Meredith Yeager[1,8], David G. Cox[17], Robert N. Hoover[1], Javed Khan[18], Gregory T. Armstrong[19], Wendy M. Leisenring[20], Smita Bhatia[21], Leslie L. Robison[19], Andreas E. Kulozik [22], Jennifer Kriebel [23,24,25], Thomas Meitinger[26,27], Markus Metzler[28], Wolfgang Hartmann[29], Konstantin Strauch[30,31], Thomas Kirchner[3,4,32], Uta Dirksen[33], Lindsay M. Morton[1], Lisa Mirabello[1], Margaret A. Tucker[1], Franck Tirode[5,6], Stephen J. Chanock[1] & Olivier Delattre[5,6]

[1]Division of Cancer Epidemiology and Genetics, National Cancer Institute, Bethesda, MD 20892, USA. [2]Max-Eder Research Group for Pediatric Sarcoma Biology, Institute of Pathology, Faculty of Medicine, LMU, 80337 Munich, Germany. [3]German Consortium for Cancer Research (DKTK), 69120 Heidelberg, Germany. [4]German Cancer Research Center (DKFZ), 69120 Heidelberg, Germany. [5]Inserm U830, Équipe Labellisés LNCC, PSL Université, Institut Curie, 75005 Paris, France. [6]SIREDO Oncology Centre, Institut Curie, 75005 Paris, France. [7]Unité de Génétique Somatique, Institut Curie, Centre Hospitalier, 75005 Paris, France. [8]Cancer Genomics Research Laboratory, Frederick National Laboratory for Cancer Research, Leidos Biomedical Research Inc, Frederick, MD 21701, USA. [9]Children's Cancer Research Institute, St. Anna Kinderkrebsforschung, 1090 Vienna, Austria. [10]Department of Oncology for Child and Adolescent, Institut Gustave Roussy, 94800 Villejuif, France. [11]Division of Pediatric Hematology Oncology and Stem Cell Transplantation, Medical Faculty, RWTH Aachen University, 52062 Aachen, Germany. [12]Human Genotyping Unit-CeGen, Human Cancer Genetics Programme, Spanish National Cancer Research Centre, 28029 Madrid, Spain. [13]Laboratorio di Oncologia Sperimentale, Istituto Ortopedico Rizzoli di Bologna, 40136 Bologna, Italy. [14]Unidad de Tumores Sólidos Infantiles, Instituto de Investigación de Enfermedades Raras, Instituto de Salud Carlos III, 28220 Majadahonda, Spain. [15]Laboratory of Pediatrics, University of Navarra, University Clinic of Navarra, IdiSNA, 31008 Pamplona, Spain. [16]Institute for Paediatric Haematology and Oncology, Leon Bérard Cancer Centre, University of Lyon, 69008 Lyon, France. [17]Centre Léon Bérard, INSERM U1052, 69008 Lyon, France. [18]Genetics Branch, Center for Cancer Research, National Cancer Institute, Bethesda, MD 20892, USA. [19]Department of Epidemiology and Cancer Control, St. Jude Children's Research Hospital, Memphis, TN 38105, USA. [20]Cancer Prevention and Clinical Statistics Programs, Fred Hutchinson Cancer Research Center, Seattle, WA 98109, USA. [21]Institute for Cancer Outcomes and Survivorship, University of Alabama, Birmingham, AL 35294, USA. [22]University Children's Hospital of Heidelberg, 69120 Heidelberg, Germany. [23]Research Unit of Molecular Epidemiology, Helmholtz Zentrum München, German Research Center for Environmental Health, 85764 Neuherberg, Germany. [24]Institute of Epidemiology, Helmholtz Zentrum München, German Research Center for Environmental Health, 85764 Neuherberg, Germany. [25]German Center for Diabetes Research (DZD), München-Neuherberg 85764, Germany. [26]Institute of Human Genetics, Helmholtz Zentrum München, German Research Center for Environmental Health, 85764 Neuherberg, Germany. [27]Institute of Human Genetics, Technische Universität München, 80333 Munich, Germany. [28]University Children's Hospital of Erlangen, 91054 Erlangen, Germany. [29]Gerhard-Domagk Institute of Pathology, University Hospital of Münster, 48149 Münster, Germany. [30]Institute of Genetic Epidemiology, Helmholtz Zentrum München, German Research Center for Environmental Health, 85764 Neuherberg, Germany. [31]Chair of Genetic Epidemiology, IBE, Faculty of Medicine, LMU, Munich 80539, Germany. [32]Institute of Pathology, Faculty of Medicine, LMU, 80337 Munich, Germany. [33]University Children's Hospital of Essen, 45147 Essen, Germany. These authors contributed equally: Stephen J. Chanock, Olivier Delattre.

