## [Peer Review File · Nature Communications]

Reviewers' comments:

Reviewer #1 (Remarks to the Author):

The manuscript from Machiela et al describes a large GWAs of Ewing sarcoma. Overall I find the data technically sound. The data provides statistical evidence for its conclusions. The results are incremental with those this group has previously published. This includes confirmation of previously identified SNPs and GGAA enrichment. The most significant problem is the lack of biological validation or attempt to understand the function of the risk variants. The data do not add to our biological understanding of Ewing beyond the purely descriptive and the authors repeatedly allude to the requirement for future studies being required for this. I do not think that this is acceptable anymore to simply catalogue SNPs and report in the absence of functional validation. The advances and "off-the-shelf" nature of validation tools - Crispr, shRNA, siRNA - make validation not only possible but necessary. In the absence of functional validation of the biological significance of some of these SNPs I do not think this study will be of significant influence to scientists in the field.

Reviewer #2 (Remarks to the Author):

The manuscript reports new results that are important for understanding the role genetic predisposition to Ewing sarcoma. The authors have undertaken an extension of their previously reported GWAS (Postal-Vinay et al. 2012) using an increased number of patients (749 vs 401). Given the rarity of the disease, this is a very substantial increase in sample size that has led to the identification of 4 new susceptibility loci at genome-wide significance.

Despite the interest of these results, there are a substantial number of issues that need to be addressed:

1. Could the authors please indicate the number of samples that were removed due to the European ancestry criterion, and explain how the ancestry was calculated? Supplementary figures with PC plots would be helpful to illustrate the sample selection and matching to controls. The authors should clarify what, if any, overlap exists between the replication samples from their 2012 study and GWAS and replication samples used in the new study.
2. For previously identified loci, Supplementary Table 1 should be extended to include the association statistics for the lead markers and any others reported in the 2012 study, so as to allow comparison between the two sets of results. Further, the authors should provide supplementary figures on association and linkage disequilibrium patterns at each of the susceptibility loci (as they have for chromosome 20), and they should provide conditional analyses of multiple markers at all of these.
3. In their previous follow-up study of the chromosome 10 locus (Grunewald et al. 2015), the primary focus was on a variant (rs79965208) that may directly impact the EWSR1-FLI1-dependent enhancer activity of the sequence by changing the length of a GGAA microsatellite. Based on the the new data reported here (lines 142-7), it seems that the disease association may be much stronger at another marker (rs10822056) on chromosome 10, but presumably this variant does not have a clear functional impact. The authors should examine this issue further with the additional analyses requested in 2 above, and discuss the impact in regards to the conclusions from the 2015 paper.
4. Although the authors present evidence that GGAA repeats within two new susceptibility loci may impact EWSR1-FLI1 binding, such a relationship appears to be widespread phenomenon associated with these microsatellite motifs but not necessarily related to genetic susceptibility. In regards to

susceptibility loci, they state that "most loci reside near GGAA repeat sequences" (line 74), but don't provide details or evaluate this observation statistically. While they state that "ChIP-seq data suggest evidence for potential interactions of germline variation ... as recently discovered at the 10q21 locus" (lines 262-4), only for the chromosome 10 locus does a disease-associated variant directly affect the microsatellite repeat. Without clear evidence linking the disease-associated variants to action of the repeat elements, the conclusion regarding the role of these repeats at several susceptibility loci as drawn here seems premature.

5. The eQTL analyses are not convincing as based on at best marginally significant p-values (Table 2) and would be best removed. Results from GTEx the relate to tissues and markers of limited relevance here (lines 218-222).

6. In the last paragraph, it is uncertain to me what conclusions the authors wish to draw about the genetic architecture of Ewing sarcoma in comparison to testicular and thyroid cancers. Similarly, I don't understand the basis for remarks about incompatibility of the effect sizes at the susceptibility loci with the lack of familial cases. The arguments in this paragraph should be clarified or dropped. The genetic risk scores (lines 245-258) provide no novel insights and these results could be summarized in single sentence rather than a longish paragraph.

Reviewer #3 (Remarks to the Author):

Overall, this is an interesting paper, but like many GWAS studies on a limited number of affected and controls, it raises more questions than it answers. The larger study reported here, building on a smaller cohort published previously, confirms 3 previously reported risk alleles and identifies 4 more. However, there are several issues to consider that deserve attention prior to publication.

1. Only 2 of the 4 new susceptibility loci replicated in independent data sets. If these are intrinsically important to the pathogenesis of ES, it is difficult to explain this failure to replicate the findings in validation sets. How do the authors explain this?

2. The 6p locus is interesting, and the associated biology studies using animal xenografts and a knock down model provide compelling data for chromatin alterations and altered gene expression at this locus, but Supplemental figure 8 offers little insight into the potential mechanism. Does the polymorphism directly affect the number of GGAA repeats? Given the importance of increasing number of GGAA repeats and binding affinity of the oncoprotein, this is an omission that should be rectified.

3. The 20p locus is also interesting, but not for the reasons noted by the authors. There are two flanking oncoprotein binding sites, embracing multiple genes. What happens to their expression in cases with susceptible polymorphisms vs. those that do not possess these polymorphisms? Given the number of ES cases for which expression data is available, a simple correlation analysis would seem to be in order.

4. The changes in expression levels for included genes in the three susceptibility loci, 6p, 8q, and 20p, after fusion gene knock down, is not very impressive, despite a claimed 85% KD. Only RREB1 and NKX2-2 show more than a 2-fold change in expression; most show minimal changes.

5. The biologic function of genes with the most marked differences in expression linked to the susceptibility loci are not cancer genes noted in COSMIC or the literature, for example. It is difficult to understand how this relates to their proposed role in oncogenesis.

6. The AUC for the aggregate of 7 susceptibility loci is indeed convincing (0.72), but the fact that unaffecteds possess 6 of the alleles on average, while ES patients possess 7 on average, is difficult to reconcile with the AUC. What is the proposed explanation?

7. In general, the paper raises some provocative issues related to genetic susceptibility, but direct links between, say, EWS-FLI1 protein binding in or near the loci and altered gene expression is less persuasive. Given the newly available on GGAA repeat binding of the oncoprotein (reference cited by

the authors), it would seem a more focused analysis of potential changes in GGAA repeat based oncoprotein binding would be indicated. For example, even an example of a polymorphism that created 4 or more GGAA repeats would be compelling evidence for a mechanism. If no such multimeric repeats are created, it would be helpful to state this.

Reviewers' comments:

Reviewer #1 (Remarks to the Author):

The manuscript from Machiela et al describes a large GWAs of Ewing sarcoma. Overall, I find the data technically sound. The data provides statistical evidence for its conclusions.

The results are incremental with those this group has previously published. This includes confirmation of previously identified SNPs and GGAA enrichment. The most significant problem is the lack of biological validation or attempt to understand the function of the risk variants. The data do not add to our biological understanding of Ewing beyond the purely descriptive and the authors repeatedly allude to the requirement for future studies being required for this. I do not think that this is acceptable anymore to simply catalogue SNPs and report in the absence of functional validation. The advances and "off-the-shelf" nature of validation tools - Crispr, shRNA, siRNA - make validation not only possible but necessary. In the absence of functional validation of the biological significance of some of these SNPs I do not think this study will be of significant influence to scientists in the field.

We thank Reviewer #1 for pointing out the high technical level of our data and for agreeing our data provides statistical evidence for its conclusions. We disagree that expanding the number of Ewing sarcoma germline susceptibility loci is incremental. The newly discovered susceptibility loci have expanded our understanding of the genetic architecture of Ewing sarcoma by providing evidence that common, high-risk germline variants may interact with the EWSR1-FLI1 translocation to alter expression of core Ewing sarcoma genes. To explore biological mechanisms important for the novel susceptibility loci, we interrogated functional relationships by analyzing chromatin markers, information on GGAA repeat sequences, EWSR1-FLI1 binding, allele specific gene expression (eQTL analyses) and knockdown of EWSR1-FLI1 in vivo. These analyses elucidated the presence of EWSR1-FLI1 binding in GGAA repeat sequences, indicates target genes with altered expression and provides evidence for alterations in target gene expression when EWSR1-FLI1 is knocked down. Further functional work to fully interrogate GGAA microsatellite regions would be substantial and require validation of the site-specific variants as well as follow-up laboratory evidence; accordingly, such effort would constitute a stand-alone paper (e.g., Grünewald et al. *Nat Genet* 2015). Moreover, the extensive work needed to validate and pursue the observations driven by "off-the-shelf" technologies is significant and could likely merit a separate paper.

Reviewer #2 (Remarks to the Author):

The manuscript reports new results that are important for understanding the role genetic predisposition to Ewing sarcoma. The authors have undertaken an extension of their previously reported GWAS (Postal-Vinay et al. 2012) using an increased number of patients (749 vs 401). Given the rarity of the disease, this is a very substantial increase in sample size that has led to the identification of 4 new susceptibility loci at genome-wide significance.

We thank Reviewer #2 for a careful review of our manuscript and appreciate the comment on our effort to analyze an expanded set of Ewing sarcoma cases.

Despite the interest of these results, there are a substantial number of issues that need to be addressed:

1. Could the authors please indicate the number of samples that were removed due to the European

ancestry criterion, and explain how the ancestry was calculated? Supplementary figures with PC plots would be helpful to illustrate the sample selection and matching to controls. The authors should clarify what, if any, overlap exists between the replication samples from their 2012 study and GWAS and replication samples used in the new study.

We calculated genetic ancestry using SNPWEIGHTS on a set of population inference SNPs (Chen et al. *Bioinformatics* 2013). The total number of Ewing sarcoma cases excluded due to less than 80% estimated European ancestry is 18 in the NCI/Curie analysis set and 5 from the CCSS analysis set. We have expanded the methods section to further describe how European ancestry was determined as well as added PCA plots of cases and controls to the Supplementary Materials (Supplementary Figure 13). Our purpose of this expanded GWAS was to include the maximum number of Ewing sarcoma cases in this analysis so we included samples from the prior 2012 GWAS replication samples. The total number of Ewing sarcoma cases that overlap between the 2012 GWAS replication set and our current analysis is 46 Ewing sarcoma cases. While extracting genotyping IDs and comparing with the prior 2012 set, we identified 16 samples incorrectly labeled as Ewing sarcoma cases that were instead relatives (e.g., parents or siblings) of Ewing sarcoma patients. These samples and the matching controls have been removed from all the analyses and all analytical results have been newly generated. The overall GWAS conclusions remain the same, with the exception of the 8q24.23 locus, which now falls below the GWAS significance threshold ($P\text{-value}=1.44\times 10^{-7}$).

2. For previously identified loci, Supplementary Table 1 should be extended to include the association statistics for the lead markers and any others reported in the 2012 study, so as to allow comparison between the two sets of results. Further, the authors should provide supplementary figures on association and linkage disequilibrium patterns at each of the susceptibility loci (as they have for chromosome 20), and they should provide conditional analyses of multiple markers at all of these.

We have extended Supplementary Table 1 to include association statistics for the lead markers from the 2012 study to allow for easy comparison between the two sets. Overall, the results between the 2012 top hits and our new top hits have a high degree of similarity. Further, we have added linkage disequilibrium plots for each susceptibility locus (Supplementary Figures 3A-G) as well as analyses conditioning on the top marker within each region (Supplementary Figures 4A-F).

3. In their previous follow-up study of the chromosome 10 locus (Grünewald et al. 2015), the primary focus was on a variant (rs79965208) that may directly impact the EWSR1-FLI1-dependent enhancer activity of the sequence by changing the length of a GGAA microsatellite. Based on the new data reported here (lines 142-7), it seems that the disease association may be much stronger at another marker (rs10822056) on chromosome 10, but presumably this variant does not have a clear functional impact. The authors should examine this issue further with the additional analyses requested in 2 above, and discuss the impact in regards to the conclusions from the 2015 paper.

Our discussion in the 2015 paper indicated that rs79965208 accounts for a fraction of the association signal at the 10q21.3 locus, but that it may not be the only SNP with regulatory effect causing the observed association signal. We believe it is plausible that other SNPs in the 10q21.3 locus could also have a regulatory effect on EGR2 expression through other mechanisms. We performed conditional analyses on the current GWAS dataset at 10q21.3 on the most highly associated variant, rs10822056 (Supplementary Table 4D). While conditioning on this variant removes most of the association signal at 10q21.3, we noted that a residual signal remains.

Interestingly, the functional variant from our 2015 paper is in stronger LD with the remaining signal after conditioning on rs10822056 compared to the reported signal (rs10822056), suggesting that more than one independent signal could be in this region. A larger sample size would be required for further confirmation. It should be noted that the chr10 locus is composed of multiple haplotype sub-blocks and multiple GGAA repeats and EWSR1-FLI1 binding sites. This suggests that the 10q21.3 region may be a complex region of Ewing sarcoma risk in which more than one functional variant probably confers risk. We have noted this in our manuscript and have accordingly added to the discussion in the 10q21.3 section of the manuscript.

4. Although the authors present evidence that GGAA repeats within two new susceptibility loci may impact EWSR1-FLI1 binding, such a relationship appears to be widespread phenomenon associated with these microsatellite motifs but not necessarily related to genetic susceptibility. In regards to susceptibility loci, they state that “most loci reside near GGAA repeat sequences” (line 74), but don’t provide details or evaluate this observation statistically. While they state that “ChIP-seq data suggest evidence for potential interactions of germline variation ... as recently discovered at the 10q21 locus” (lines 262-4), only for the chromosome 10 locus does a disease-associated variant directly affect the microsatellite repeat. Without clear evidence linking the disease-associated variants to action of the repeat elements, the conclusion regarding the role of these repeats at several susceptibility loci as drawn here seems premature.

We agree with Reviewer #2 that evidence is needed to link Ewing sarcoma susceptibility loci to variation in GGAA microsatellites. To evaluate the observation that most loci reside near EWS-FLI1 binding sequences statistically, we have investigated the vicinity of Ewing sarcoma susceptibility hits with EWSR1-FLI1 ChIP-seq binding peaks in A673 and TC71 cell lines (Supplementary Table 4, Supplementary Figure 8). We find SNPs in Ewing sarcoma susceptibility loci are on average significantly closer to EWSR1-FLI1 bound elements than would be expected by chance on a chromosome-wide level (Wilcoxon P-values= 0.0025 and 0.0009 in A673 and TC71 cell lines, respectively). These findings indicate enrichment of EWSR1-FLI1 binding near Ewing sarcoma susceptibility regions and provide further evidence suggesting a link between germline susceptibility variants and EWSR1-FLI1 response elements.

5. The eQTL analyses are not convincing as based on at best marginally significant p-values (Table 2) and would be best removed. Results from GTEx the relate to tissues and markers of limited relevance here (lines 218-222).

We agree with Reviewer #2 that, while statistically significant, the reported eQTL analyses do not provide conclusive support for allele specific differences in gene expression in these susceptibility regions. The limited statistical support for eQTLs in this region is primarily due to the small sample size of gene expression sample sets to produce a well-powered analysis. Interestingly, for the regions we identified eQTLs, we find biologically relevant genes which we feel justifies the inclusion of these eQTL analyses in the manuscript. In particular for *RREB1*, we find altered expression levels with knock down of EWSR1-FLI1. In response to Reviewer #2’s concern regarding the preliminary nature of these eQTL results, we have shortened these sections in the manuscript while further investigations are carried out with respect to the functional effects of these GWAS susceptibility signals. We have also added Supplementary Table 6 which explores eQTLs with all genes in the 20p11.22 and 20p11.23 regions in GTEx samples. These analyses show eQTLs with *KIZ* in many tissue types, but no reported eQTLs for *NKX2-2*. This replicates the *KIZ* eQTL in other tissues and suggests it is not observed only in Ewing

sarcoma tumors.

6. In the last paragraph, it is uncertain to me what conclusions the authors wish to draw about the genetic architecture of Ewing sarcoma in comparison to testicular and thyroid cancers. Similarly, I don't understand the basis for remarks about incompatibility of the effect sizes at the susceptibility loci with the lack of familial cases. The arguments in this paragraph should be clarified or dropped. The genetic risk scores (lines 245-258) provide no novel insights and these results could be summarized in single sentence rather than a longish paragraph.

The comparison of Ewing sarcoma with testicular and thyroid cancer was intended to highlight the unexpected high number of our Ewing sarcoma GWAS signals with elevated odds ratios based on so few cases, suggesting the discovery to case ratio is particularly high. As suggested by Reviewer #2, we clarified this section by removing the remarks about heritability and family studies as well as limited the discussion of the genetic risk score.

Reviewer #3 (Remarks to the Author):

Overall, this is an interesting paper, but like many GWAS studies on a limited number of affected and controls, it raises more questions than it answers. The larger study reported here, building on a smaller cohort published previously, confirms 3 previously reported risk alleles and identifies 4 more. However, there are several issues to consider that deserve attention prior to publication.

We thank Reviewer #3 for her/his review of our manuscript. Below are responses to the points raised.

1. Only 2 of the 4 new susceptibility loci replicated in independent data sets. If these are intrinsically important to the pathogenesis of ES, it is difficult to explain this failure to replicate the findings in validation sets. How do the authors explain this?

The limited sample size of the replications sets can result in inadequate power to detect susceptibility variants detected in the larger GWAS meta-analysis. The failure of a genome-wide significant variant to replicate in a smaller study is a common feature in adult and pediatric cancer GWAS (Thomas et al. Nature Genetics 2008, Michailidou et al. Nature 2017) and does not communicate information about the intrinsic importance of a variant in the pathogenesis of a cancer. In fact, our analysis shows a high degree of consistency of effect across study for both the previously published and the new Ewing sarcoma susceptibility loci (Supplementary Figures 2 and 5).

2. The 6p locus is interesting, and the associated biology studies using animal xenografts and a knock down model provide compelling data for chromatin alterations and altered gene expression at this locus, but Supplemental figure 8 offers little insight into the potential mechanism. Does the polymorphism directly affect the number of GGAA repeats? Given the importance of increasing number of GGAA repeats and binding affinity of the oncoprotein, this is an omission that should be rectified.

We believe that germline genetic variation at the Ewing sarcoma susceptibility locus is affecting binding of EWSR1-FLI1 at nearby GGAA repeat sequences. The mechanism responsible for this relationship is unclear, however, we hypothesize the GWAS variant could be tagging variation in an

EWSR1-FLI1 binding site, in particular the length or structure of GGAA repeats in a polymorphic GGAA microsatellite (as in Grünewald et al. *Nat Genet* 2015), or in some way altering nearby chromatin structure and affecting EWSR1-FLI1 binding. Unfortunately, developing targeted sequencing assays to sequence repetitive, low-complexity regions and detecting nearby variation is complex and will require significant time before we can disentangle the mechanisms linking these GWAS susceptibility signals to altered EWSR1-FLI1 binding. To address some of Reviewer #3's concerns, we have investigated the effects of correlated variants in LD (drawn from dbSNP and using either R² or D') with the lead GWAS Ewing sarcoma susceptibility variants, particularly in the most proximal GGAA microsatellite (Supplementary Table 5); specifically, we looked at the effect of correlated SNPs on microsatellite length. Accordingly, we have added the following text to the manuscript: "Several variants correlated with rs7742053 are in contiguity with the GGAA repeat and may be candidate functional variants that disrupt EWSR1-FLI1 binding (Supplementary Table 5). One such variant, rs10541084, a -/GAAG indel is located at the telomeric end of the nearest GGAA microsatellite and is in LD with rs7742053 (R²_{CEU}=0.15, D'_{CEU}=0.92). Interestingly, the rs7742053 risk A allele is correlated with the rs10541084 GAAG allele which is more common in Europeans, extends the microsatellite GGAA repeat sequence, and may improve binding of EWSR1-FLI1. This evidence suggests a similar mechanism as in the 10q21 locus may be acting at the 6p25.1 locus in which variation in a GGAA repeat affects EWSR1-FLI1 binding leading to altered expression of *RREB1* or an alternative nearby gene. Further functional work at 6p25.1 is required to clarify which variants are functionally responsible for the susceptibility signal."

3. The 20p locus is also interesting, but not for the reasons noted by the authors. There are two flanking oncoprotein binding sites, embracing multiple genes. What happens to their expression in cases with susceptible polymorphisms vs. those that do not possess these polymorphisms? Given the number of ES cases for which expression data is available, a simple correlation analysis would seem to be in order.

As Reviewer #3 has noted, there are many genes in the 20p11.22-23 region. When we performed our eQTL analyses, we investigated all associations with proxies of the Ewing sarcoma association signals with expression differences in all genes in the region. As reported in our manuscript, we see no evidence for an eQTL with *NKX2-2*, but see evidence for allele specific differences in expression of *KIZ*. To further clarify in the manuscript that we investigated for all potential eQTLs in the region we have added Supplementary Table 6, which shows all genes tested and whether there was evidence for an association (P<0.05).

4. The changes in expression levels for included genes in the three susceptibility loci, 6p, 8q, and 20p, after fusion gene knock down, is not very impressive, despite a claimed 85% KD. Only *RREB1* and *NKX2-2* show more than a 2-fold change in expression; most show minimal changes.

To highlight potential candidate genes in the Ewing sarcoma susceptibility regions, we investigated the effect of gene expression on all genes located at each Ewing sarcoma locus when EWSR1-FLI1 was knocked down. As Reviewer #3 noted, almost all genes at the corresponding loci showed little change in expression when EWSR1-FLI1 was knocked down, indicating these genes are not (de)regulated by EWSR1-FLI1. The exception was for *RREB1* and *NKX2-2*, which show notable changes in expression when EWSR1-FLI1 is knocked down, suggesting these are important candidate target genes in the Ewing sarcoma susceptibility regions. The specific regulation of *RREB1* and *NKX2-2* as compared to all other genes located in the given susceptibility locus, underscores their potential roles as candidate genes, which, in turn merit further follow-up.

5. The biologic function of genes with the most marked differences in expression linked to the susceptibility loci are not cancer genes noted in COSMIC or the literature, for example. It is difficult to understand how this relates to their proposed role in oncogenesis.

Not all genes important to cancer risk are oncogenic. Several genes are transformation facilitating genes, which are not sufficient for transformation, but dysregulate core pathways and are not usually mutated in cancer. A recent analysis of cancer genome-wide significant loci found nearby genes regulated by GWAS loci are not preferential targets for somatic mutation (Machiela et al. Genome Biology 2015). This highlights why GWAS is a complementary tool to identify important genes in cancer etiology, especially in cancers with few somatic mutations such as Ewing sarcoma.

6. The AUC for the aggregate of 7 susceptibility loci is indeed convincing (0.72), but the fact that unaffecteds possess 6 of the alleles on average, while ES patients possess 7 on average, is difficult to reconcile with the AUC. What is the proposed explanation?

We appreciate Reviewer #3 highlighting a point of confusion. Most variants identified in Ewing sarcoma susceptibility regions are common, so one would expect these variants to be common in controls as well. While the AUC of 0.72 is impressive, the AUC is inflated because the genetic risk score was tested in the discovery set. An independent sample would provide a more accurate measure of discriminative ability. In response to Reviewer #3's comments, we have reduced the discussion on the genetic risk score, removed AUC from the discussion and emphasized that population based screening of these variants would be ineffective.

7. In general, the paper raises some provocative issues related to genetic susceptibility, but direct links between, say, EWS-FLI1 protein binding in or near the loci and altered gene expression is less persuasive. Given the newly available on GGAA repeat binding of the oncoprotein (reference cited by the authors), it would seem a more focused analysis of potential changes in GGAA repeat based oncoprotein binding would be indicated. For example, even an example of a polymorphism that created 4 or more GGAA repeats would be compelling evidence for a mechanism. If no such multimeric repeats are created, it would be helpful to state this.

We agree with Reviewer #3 that providing a direct link between the Ewing sarcoma GWAS susceptibility variants and GGAA repeat binding of the EWSR1-FLI1 oncoprotein would add to the manuscript. In response to Reviewer #2's point #4, we carried out an analysis of EWSR1-FLI1 binding near Ewing sarcoma GWAS susceptibility regions and found significant evidence for enrichment of EWSR1-FLI1 binding near Ewing sarcoma susceptibility regions. In addition, we added an example of a variant at 6p25.1 in LD with the GWAS variant and that alters GGAA microsatellite length or structure (see Reviewer #3 point #2). Overall, it is technically difficult to conduct long-range sequence analysis of the GGAA repeat sequences for large numbers of Ewing sarcoma cases to identify polymorphic GGAA repeats or variants that interrupt the repeat sequences. Such an effort would warrant a new study centered on a detailed genomic analysis on the association of susceptibility loci with regions of variable length GGAA repeats.

Reviewers' comments:

Reviewer #1 (Remarks to the Author):

The authors have modified the manuscript in response to the comments of the reviewers which has clarified the text.

Whilst I acknowledge the comments from the authors, I remain unconvinced that a study that is purely GWAS and statistical association demonstrating (essentially) a modest advance over previous work from the same group is appropriate for this forum (this is an editorial decision). Biological validation should not be considered secondary to the GWAS.

Reviewer #2 (Remarks to the Author):

The manuscript is significantly improved over the previous version, and I will make no further comments regarding points raised in my first review. However, the revision contains new observations on the impact of a possible disease-associated marker affecting a GGAA repeat at the 6p25.1 locus. While I accept that the authors may not be able to clarify the potential effects of this variant on EWSR1-FLI1 binding fully in this manuscript, they could provide additional details regarding the association of the -/GAAG indel (rs10541084) with the disease. If it is not possible to genotype rs10541084 - which would be the preferred option - more information might be obtained through imputation and conditional statistical analysis.

Reviewer #2 comments on Reviewer #3 rebuttal (Remarks to the Authors)

- With respect to the previous query regarding whether disease-associated polymorphisms at the 6p locus directly affect the number of GGAA repeats, the authors have added some markers in LD with the lead SNP. This is an interesting new observation. The authors say that they cannot follow-up this from a mechanistic viewpoint here (which I can accept) but they still have not provided key information to know if this variant accounts for the association.
- The authors have included an additional table (Supplementary Table 6) showing information about potential eQTL effects from ES and GTEx data for genes within the chromosome 20 disease-associated loci. However, interpretation of the information in Supplementary Table 6 is not clear: the color code in the table confounds all results with (nominal) $p < 0.05$ allowing no assessment of the strength of the eQTL association; furthermore, there is no indication of the strength of the relationship with disease-associated markers as all SNPs with $r^2 > 0.1$ are included without distinguishing between those that are in strong LD with the lead variants. Since the authors identify a (or the) primary candidate for the 20p11.22 locus as NKX2-2 based on the results from the fusion KD, and this gene does not show eQTL effects, I think that the authors need to clarify further Supplementary Table 6, and comment appropriately in the text (around lines 222-223) on the strength of the eQTL associations which don't seem to be pointing to the likely candidate.
- Regarding the reduced discussion on genetic risk, the authors now state "that population based screening using these six variants would be ineffective at this time". I'm happy with this response, however I think rather than saying "this time" it would be better to say "unlikely to be effective".

Reviewers' comments:

Reviewer #1 (Remarks to the Author):

The authors have modified the manuscript in response to the comments of the reviewers which has clarified the text.

We thank Reviewer #1 for their review of our revised manuscript and are pleased they find the revisions have clarified the text.

Whilst I acknowledge the comments from the authors, I remain unconvinced that a study that is purely GWAS and statistical association demonstrating (essentially) a modest advance over previous work from the same group is appropriate for this forum (this is an editorial decision). Biological validation should not be considered secondary to the GWAS.

We do not agree with Reviewer #1's comment that our manuscript represents a modest advance over previous work. For a rare, pediatric tumor, substantial work was required to amass a larger sample size for the GWAS meta-analysis. Our meta-analysis has identified 3 new independent signals for Ewing susceptibility, suggests new target genes, and builds on prior evidence suggesting further interactions could exist between germline variation in or around GGAA repeat regions and *EWSR1-FLII* binding. This manuscript represents an important step in understanding the genetic etiology of Ewing sarcoma and identified risk alleles with estimated effect sizes larger than those normally observed in adult cancer GWAS. Functional work is complicated in GGAA repeat regions, requires significant time and resources, and is beyond the scope of this manuscript.

Reviewer #2 (Remarks to the Author):

The manuscript is significantly improved over the previous version, and I will make no further comments regarding points raised in my first review.

We thank Reviewer #2 for reviewing our revision and appreciate the kind words indicating the manuscript is significantly improved.

However, the revision contains new observations on the impact of a possible disease-associated marker affecting a GGAA repeat at the 6p25.1 locus. While I

accept that the authors may not be able to clarify the potential effects of this variant on EWSR1-FLI1 binding fully in this manuscript, they could provide additional details regarding the association of the -/GAAG indel (rs10541084) with the disease. If it is not possible to genotype rs10541084 - which would be the preferred option - more information might be obtained through imputation and conditional statistical analysis.

As suggested by Reviewer #2, herein we provide additional details regarding the association of rs10541084 with EWS risk. rs10541084 had an association p-value of 0.01 in the overall meta-analysis. The indel was well-imputed in all 3 analysis sets (info scores 0.85, 0.93, and 0.98), suggesting direct genotyping would likely have little effect on the association with EWS risk. rs10541084 is in LD with the chr6 lead SNP (rs7742053) with $R^2_{CEU}=0.15$ and $D'_{CEU}=0.92$ and the risk A allele of rs7742053 corresponds to the microsatellite lengthening GAAG allele of rs105441084. This indel likely doesn't account for the entire observed association at 6p25.1, suggesting a complex interaction between germline variation in EWSR1-FLI1 binding where additional variation in or around the GGAA microsatellite could have an impact on EWSR1-FLI1 binding. We have modified the text of the manuscript to include information on rs10541084. The text now reads:

“Several variants correlated with rs7742053 are in contiguity with the GGAA repeat and may be candidate functional variants that disrupt EWSR1-FLI1 binding (Supplementary Table 5). One such variant, rs10541084, a -/GAAG indel is located at the telomeric end of the nearest GGAA microsatellite, is in LD with rs7742053 ($R^2_{CEU}=0.15$, $D'_{CEU}=0.92$)²⁵, and is nominally associated with EWS (OR=1.20, 95% CI=1.04-1.37, P-value=0.01). Interestingly, the rs7742053 risk A allele is correlated with the rs10541084 GAAG allele which is more common in Europeans, extends the microsatellite GGAA repeat sequence, and could enhance binding of EWSR1-FLI1. This evidence suggests that a similar mechanism as in the 10q21 locus²³ may be acting at the 6p25.1 locus in which variation of a GGAA repeat affects EWSR1-FLI1 binding leading to altered expression of *RREB1* or an alternative nearby gene. Further functional work at 6p25.1 is required to clarify which variants are functionally responsible for the susceptibility signal.”

Reviewer #2 comments on Reviewer #3 rebuttal (Remarks to the Authors)

We thank Reviewer #2 for responding to Reviewer #3's comments to expedite the review process.

- With respect to the previous query regarding whether disease-associated polymorphisms at the 6p locus directly affect the number of GGAA repeats, the authors have added some markers in LD with the lead SNP. This is an interesting new observation. The authors say that they cannot follow-up this from a mechanistic viewpoint here (which I can accept) but they still have not provided key information to know if this variant accounts for the association.

We agree with Reviewer #2 that the 6p25.1 locus is indeed interesting for EWS susceptibility, particularly since *RREB1* is a nearby candidate gene with evidence for allele specific expression differences (P-value=0.01). We refer Reviewer #2 back to the response above (Response to Reviewer #2 Remarks to the Author) with respect to the association of one indel (rs10541084) in a GGAA repeat sequence near 6p25.1 and EWS risk. While this variant may account for some of the associated risk tagged by rs7742053 at the 6p25.1 locus, the modest meta-analysis association P-value suggests there is likely additional variation in the region that also impacts EWSR1-FLI1 binding and dysregulation of nearby target genes. Similar observations in our prior work functionally mapped the 10q21 locus (Grünwald *et al. Nat Genet* 2015) in which the GWAS marker rs224278 had a lower association p-value than the putatively functional variant rs79965208 (P-value 4.0×10^{-17} versus 0.022); however, independent replication sets gave additional support to the rs79965208 variant (P-value= 6.15×10^{-3} and P-value= 9.33×10^{-6}). These associations suggest GWAS identified regions of EWS susceptibility are complex and EWSR1-FLI1 binding in these regions is likely governed by multiple inherited variants in the region. Extensive long-range, haplotype-based sequencing in EWS cases and cancer-free controls, although beyond the scope of our manuscript, is needed to better resolve how inherited variation in the 6p25.1 region is associated to EWS susceptibility.

- The authors have included an additional table (Supplementary Table 6) showing information about potential eQTL effects from ES and GTEx data for genes within the chromosome 20 disease-associated loci. However, interpretation of the information in Supplementary Table 6 is not clear: the color code in the table confounds all results with (nominal) $p < 0.05$ allowing no assessment of the strength of the eQTL association; furthermore, there is no indication of the strength of the relationship with disease-associated markers as all SNPs with $r^2 > 0.1$ are included without distinguishing between those that are in strong LD with the lead variants. Since the authors identify a (or the) primary candidate for the 20p11.22 locus as NKX2-2 based on the results from the fusion KD, and this gene does not show

eQTL effects, I think that the authors need to clarify further Supplementary Table 6, and comment appropriately in the text (around lines 222-223) on the strength of the eQTL associations which don't seem to be pointing to the likely candidate.

We agree with Reviewer #2 that Supplementary Table 6 could be improved with additional information and editing to clearly communicate our point. We have accordingly modified the table to only include variants in moderate to high linkage disequilibrium ($R^2_{CEU}>0.50$) and have indicated the level of linkage disequilibrium for each surrogate SNP in the table. We have also removed the color scheme and have directly reported the eQTL p-values from GETx. We believe that the new table is more informative, directly conveying our salient points, namely that eQTLs with *NKX2-2* are not observed in other tissues (as was also observed for EWS tumors) and eQTLs with *KIZ* (GTEx alias=*PLK1S1*) are observed in artery, sun exposed skin, testis and whole blood (as was observed for EWS tumors). The revised Supplementary Table 6 is below.

GTEx Tissue	EWS GWAS Region	Most Associated Variant (R^2 CEU)	eQTL Association P-value by Gene					
			INSM1	LINC00261	NKX2-2	PAX1	KIZ*	XRN2
Artery - Tibial	20p11.22		n.s.	n.s.	n.s.	n.s.	n.s.	n.s.
	20p11.23	rs11087340 (1.00)	n.s.	n.s.	n.s.	n.s.	8.0E-05	n.s.
Cells - Transformed fibroblasts	20p11.22	rs112337001 (0.51)	n.s.	n.s.	n.s.	n.s.	n.s.	7.3E-06
	20p11.23		n.s.	n.s.	n.s.	n.s.	n.s.	n.s.
Skin - Sun Exposed (Lower leg)	20p11.22	rs6035892 (0.54)	n.s.	n.s.	n.s.	n.s.	4.2E-07	n.s.
	20p11.23	rs6047244 (0.72)	n.s.	n.s.	n.s.	n.s.	1.8E-08	n.s.
Testis	20p11.22	rs6035892 (0.54)	n.s.	n.s.	n.s.	n.s.	4.2E-06	n.s.
	20p11.23	rs6137252 (1.00)	n.s.	n.s.	n.s.	n.s.	5.1E-10	n.s.
Whole Blood	20p11.22		n.s.	n.s.	n.s.	n.s.	n.s.	n.s.
	20p11.23	rs6137246 (0.96)	n.s.	n.s.	n.s.	n.s.	1.4E-11	n.s.

**KIZ* is denoted as the alias *PLK1S1* in GETx

n.s.=not significant

We have also clarified the associated text in the manuscript. The relevant sections now read:

“We explored eQTLs for other tissue types in GTEx with surrogate SNPs in moderate to high linkage disequilibrium with rs6047482, but found no evidence for an eQTL with *NKX2-2* in these tissues likely due to EWS specific expression of *NKX2-2* (Supplementary Table 6)²⁹. It is plausible that EWSR1-FLI1-induced elevated *NKX2-2* expression levels in EWS cells hamper our ability to detect allele specific expression patterns of *NKX2-2* that may be important for EWS transformation in the EWS progenitor cells. Further

eQTL analyses in a large set of MSCs, the suspected EWS cell-of-origin, should enable this hypothesis to be tested. ”

“This eQTL at 20p11.23 association with KIZ does not appear to be restricted to EWS and was observed in other GTEx tissues (e.g., artery, sun-exposed skin, testis and whole blood; Supplementary Table 6).”

- Regarding the reduced discussion on genetic risk, the authors now state "that population based screening using these six variants would be ineffective at this time". I'm happy with this response, however I think rather than saying "this time" it would be better to say "unlikely to be effective".

At Reviewer #2’s suggestion, we have revised the sentence to now read:

“Due to the rarity of EWS and the relatively high frequency of these common susceptibility alleles, absolute risks of EWS associated with these six EWS susceptibility loci are low suggesting population based screening using these six variants is unlikely to be effective.”

REVIEWERS' COMMENTS:

Reviewer #1 (Remarks to the Author):

I have no additional comments to add. My comments were submitted with the previous review and the authors have provided their argument that biological validation is unnecessary. As mentioned previously, the handling editor should decide on the available evidence, reviews and the responses to these that have been already received.

No separate/additional comments were provided to the editor.